# Impact of commonly administered drugs on the progression of spinal cord injury: a systematic review

Lucie Bourguignon [1,2,3,10] ✉, Louis P. Lukas [1,2,3,10] ✉, Bethany R. Kondiles [4,5], Bobo Tong[4], Jaimie J. Lee [4], Tomás Gomes[6], Wolfram Tetzlaff[4,5], John L. K. Kramer[4,7,8], Matthias Walter [9] & Catherine R. Jutzeler[1,2,3]

## Abstract

**Background** Complications arising from acute traumatic spinal cord injury (SCI) are routinely managed by various pharmacological interventions. Despite decades of clinical application, the potential impact on neurological recovery has been largely overlooked. This study aims to highlight commonly administered drugs with potential disease-modifying effects.

**Methods** This systematic literature review included studies referenced in PubMed, Scopus and Web of Science from inception to March 31st, 2021, which assess disease-modifying properties on neurological and/or functional recovery of drugs routinely administered following spinal cord injury. Drug effects were classified as positive, negative, mixed, no effect, or not (statistically) reported. Risk of bias was assessed separately for animal, randomized clinical trials, and observational human studies.

**Results** We analyzed 394 studies conducting 486 experiments that evaluated 144 unique or combinations of drugs. 195 of the 464 experiments conducted on animals (42%) and one study in humans demonstrate positive disease-modifying properties on neurological and/or functional outcomes. Methylprednisolone, melatonin, estradiol, and atorvastatin are the most common drugs associated with positive effects. Two studies on morphine and ethanol report negative effects on recovery.

**Conclusion** Despite a large heterogeneity observed in study protocols, research from bed to bench and back to bedside provides an alternative approach to identify new candidate drugs in the context of SCI. Future research in human populations is warranted to determine if introducing drugs like melatonin, estradiol, or atorvastatin would contribute to enhancing neurological outcomes after acute SCI.

## Plain language summary

Patients with spinal cord injury (SCI) are exposed to a wide range of medications treating health conditions arising as a consequence of the initial injury. The effect of providing patients with a large number of medications in the early period after injury, that is in the first days to weeks, on recovery from SCI, however, is typically not considered. This extensive and structured review of evidence from pre-clinical (animal) and clinical (human) studies quantifies these effects for the first time. 144 unique drugs or combinations of drugs previously reported to be administered in animal models or to patients with SCI have been studied for their effect on recovery across 486 distinct experiments. A small subset of drugs are associated with positive effects, and provide potential targets for further study to determine if they can be used to treat SCI.

Spinal cord injury (SCI) is a devastating condition that often leads to severe and permanent neurological and functional impairments. Despite recent advancements, effective treatments promoting neurological and functional recovery are urgently needed[1,2]. Over the last decades, interest in exploring the disease-modifying effects of commonly administered drugs in this context has grown[3–6]. Nearly every individual sustaining a traumatic SCI receives multiple types and classes of drugs to manage a wide range of secondary complications associated with the neurotrauma[7–9]. These range

[1]Department of Health Sciences and Technology, ETH Zurich, Zurich, Switzerland. [2]SIB Swiss Institute of Bioinformatics, Lausanne, Switzerland. [3]Schulthess Klinik, Zurich, Switzerland. [4]International Collaboration on Repair Discoveries (ICORD), University of British Columbia, Vancouver, BC, Canada. [5]Department of Zoology, University of British Columbia, Vancouver, BC, Canada. [6]Department of Biosystems Science and Engineering, ETH Zurich, Basel, Switzerland. [7]Djavad Mowafaghian Centre for Brain Health, University of British Columbia, Vancouver, BC, Canada. [8]Department of Anesthesiology, Pharmacology, and Therapeutics, Faculty of Medicine, University of British Columbia, Vancouver, BC, Canada. [9]Department of Urology, University Hospital Basel, University of Basel, Basel, Switzerland. [10]These authors contributed equally: Lucie Bourguignon, Louis P. Lukas. ✉e-mail: lucie.bourguignon@etu.u-bordeaux.fr; louis.lukas@hest.ethz.ch

from drugs to manage blood pressure, to analgesics for concomitant traumatic injuries, to anticholinergics for spasms. A recent study showed that patients receive up to 60 unique drugs within the first 2 months, often in combinatorial fashion[7]. Despite extensive polypharmacy, little is known to what degree drugs commonly used in the management of acute SCI have downstream, unintended, beneficial or detrimental, effects on neurological and functional outcomes.

The acute phase of SCI represents a crucial window of opportunity for therapeutic intervention. Consequently, understanding the potential therapeutic benefits or possible harm of routinely administered drugs on neurological and functional recovery is paramount in the development of effective treatment strategies for SCI. The detrimental effects of SCI extend beyond the initial damage, as a cascade of secondary injury processes like inflammation, oxidative stress, excitotoxicity, and apoptosis is triggered further compromising neural tissue and impeding recovery. Identifying drugs that can modify these secondary injury mechanisms while promoting neural repair and regeneration presents a promising avenue of research. Commonly administered drugs, already approved for various medical conditions, offer the advantage of established safety profiles and known pharmacokinetics. These drugs have been extensively studied in their primary therapeutic indications, but emerging evidence suggests that some possess additional neuroprotective, neuroregenerative, or anti-inflammatory properties potentially promoting recovery after SCI[6, 10]. Disease-modifying effects of these drugs can be multifaceted. Some drugs may act directly on the injured spinal cord by reducing inflammation[11], inhibiting cell death pathways[12], or promoting axonal regeneration[13]. Others may exert their effects indirectly by modulating the surrounding environment, such as promoting angiogenesis or altering the immune response[14, 15] to create a more conducive environment for neural repair. Simultaneously, potential harmful effects of commonly administered drugs on neurological recovery are rarely considered but their identification could allow for crucial changes in treatment strategies.

To bridge this knowledge gap, we conducted a comprehensive systematic review of preclinical and clinical studies examining the effects of commonly administered drugs on functional and neurological recovery following SCI. We found extensive heterogeneity across study parameters, which could potentially complicate translation of promising findings from preclinical studies but also highlights opportunities for further investigation of promising candidates for drug repurposing.

## Methods
The study protocol was registered with and approved by the international prospective register of systematic reviews (PROSPERO) (registration number: CRD42021231851). This review conforms to the Preferred Reporting Items for Systematic reviews and Meta-Analysis (PRISMA) guidelines[16].

### Selection of drugs
The list of all drugs administered in the first 60 days after injury to treat secondary complications in the Sygen[17] and SCIRehab[18] cohorts were extracted from our recent publication[7]. We will refer to those as drugs of interest. The subset of drugs for which studies could be retrieved and were included in this review is provided in Supplementary Data S1.

### Search methods for identification of studies
Using Publish or Perish (version 7.23.2852.7498)[19], PubMed, SCOPUS, and Web of Science were searched using the time range from their individual inception dates (1977, 1960, and 1945 respectively) to March 31st, 2021. Search terms were *spinal cord injury*, *recovery*, and *name* of a drug of interest (Section "Selection of drugs"), joined with AND. A manual search was also performed to include matching references of relevant trials.

### Selection of studies
Articles were independently screened in two stages: initial screening of titles and abstracts (MW, CRJ), and full-text assessments (LB, LPL, MW, CRJ)

using criteria described in the following section. In case multiple articles reported on a single cohort, the article providing the most data or detail was selected for further synthesis. Disagreements were discussed and resolved at multiple consensus meetings.

### Inclusion and exclusion criteria
All full-text, peer-reviewed studies investigating the disease-modifying effect of a drug of interest on relevant neurological or functional outcomes after acute SCI were included. Where original articles were not published in English, screening and data extraction were performed by native speakers. We excluded duplicates, non peer-reviewed articles, reviews, meta-analyses, abstracts, editorials, commentaries, perspectives, patents, letters with insufficient data reporting, studies exclusively on children/neonates, or out-of-scope studies (see Fig. 1 for full definition). In particular, out-of-scope studies included publications investigating drugs outside of the drugs of interest as defined in *Selection of Drugs*. We only included studies comparing the treatment group to a placebo control group, and excluded experiments using active compounds as the only control as it is impossible to compare drug effects between studies using different comparators (i.e., different active controls in studies A and B instead of placebo). Authors of articles that were indexed but not accessible either through institutional library access (ETH Zurich) or open source publishing, were contacted to obtain a copy of the full article. In case no copy was provided, the article was excluded (see *not accessible* in Fig. 1). Subsequent data extraction was performed by six investigators (LB, LPL, BT, JL, TG, and CRJ).

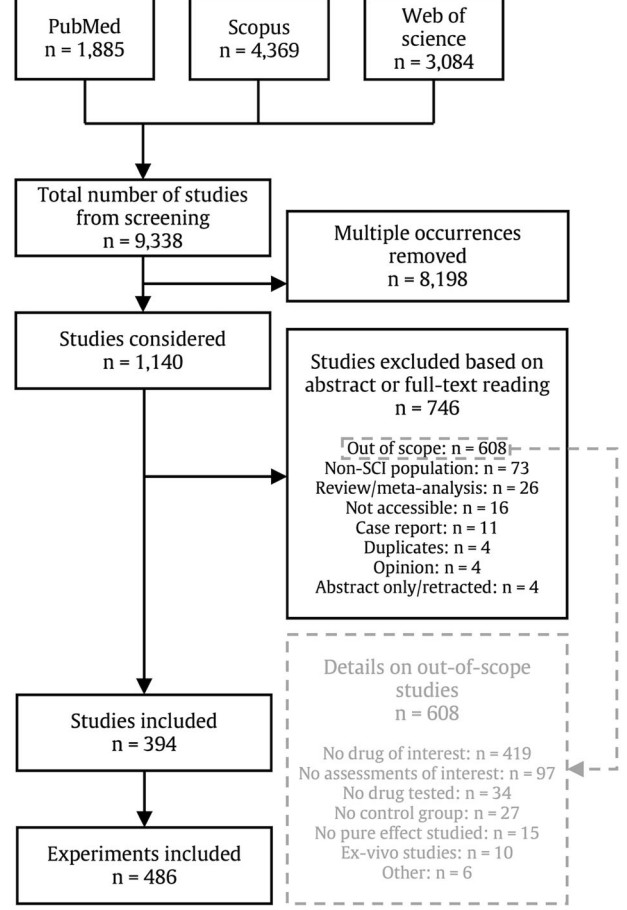

**Fig. 1 | PRISMA flowchart.** Protocols (*n* = 4), non-standardized spinal cord injury model (*n* = 1), and capsaicin-based TRPV1 study (*n* = 1) are grouped under *other* out-of-scope excluded studies.

**Table 1 | Classification of drug effect**

| Drug Effect | Description |
|---|---|
| Positive | Treatment with the drug of interest resulted in improved/increased functional/neurological outcomes compared to control. |
| Negative | Treatment with the drug of interest resulted in worse/decreased functional/neurological outcomes compared to control. |
| No effect | Treatment with the drug of interest did not impact the functional/neurological outcomes compared to control in a statistically significant manner. |
| No statistics | Qualitative comparison between treatment and control groups were performed, but no statistical test results were reported. |
| Not reported | Functional/neurological outcomes were defined in Methods but results of comparison between treatment and control groups were not reported. |
| Mixed | Combination of positive, negative, no effects and/or no statistics was reported, depending on the assessments, dosage, timing, regimen or a combination of those situations. |

## Assessments and outcomes

The review focused on studies reporting drug effects on recovery as assessed by locomotor function, skilled fore- or upper limb function, sensory function as well as electrophysiology. Details about the assessments included in the analysis are reported in Supplementary Data S2 and Supplementary Table S1. Assessments used to track recovery outcomes in animals with SCI were grouped into categories based on the deficits measured. Tasks that assess spontaneous and voluntary motor function were differentiated between quadrupedal locomotion or skilled reaching or forelimb usage. Sensory assessments were grouped, including sensory reflex arcs, regardless of the type of sensory input eliciting the reflex. Assessments of electrical activity of muscle fibers or circuits were grouped under electrophysiology assessments to mirror comparable assessments in humans and reflect neural excitability. Too few papers assessed reflexes or utilized electrophysiology to warrant distinguishing between proprioceptive or pain/withdrawal reflexes, or between assessments of single units vs. monosynaptic or polysynaptic potentials or motor vs. sensory circuits. Assessments spanning multiple categories (e.g., Gale scale) or used in only a few studies were grouped together. In cases of ambiguity, the methods and results of the paper were closely reviewed to ascertain the feature of the deficit being assessed (e.g., toe spread as a measure of reflexes vs. weight bearing during locomotion).

## Data extraction and synthesis

The following information was extracted from all studies: (1) study characteristics (first author's last name, publication year, language), (2) study population (species, group sizes [total/control/treatment], sex, age, weight), (3) injury characteristics (level, severity, mechanism, duration), (4) drug administration (drug name, dose, route of administration, timing of start of treatment relative to injury, duration of treatment), and (5) neurological and functional assessment outcomes (name, time point(s), investigators blinded to treatment, drug effect). A full list of extracted variables is provided in Supplementary Data S3. Studies analyzing multiple drugs of interest (e.g., drug A, drug B, and control, with drugs A and B of interest) were separated into multiple experiments (e.g., experiment 1: drug A vs. control, experiment 2: drug B vs. control) and extracted individually. Clinical studies on human populations were assessed for risk of bias (RoB) according to their design, either using the RoB 2 tool for randomized clinical trials (RCTs)[20] or the ROBINS-I tool for non-randomized interventions[21]. Animal experiments were assessed for risk of bias based on the SYstematic Review Centre for Laboratory animal Experimentation (SYRCLE) RoB tool[22]. Additionally, incomplete reporting of basic information relating to the study protocol was graded with a score from 0 (no selective reporting) to 20 (highest selective reporting) according to criteria listed in Supplementary Table S2. Visualizations for RoB assessments of RCTs and intervention studies were created using *robvis*[23].

## Statistical analysis

Drug effects were classified for each experiment in one of six categories (Table 1). Descriptive statistics (mean, standard deviations, median, min, max, percentage, and proportions) were used to provide summary information on the study characteristics, the studied drugs and their effect on recovery after SCI.

## Reporting summary

Further information on research design is available in the Nature Portfolio Reporting Summary linked to this article.

## Results

Initially 9338 studies were screened and 1140 qualified for full-text reading. 394 unique studies, reporting 486 experiments, met our inclusion criteria (Fig. 1). Sixty-four studies (16%) reported more than one experiment. Studies were published between June 1975 and March 2021, with the majority after 2010 (238 studies, 60%, Supplementary Fig. S1A). While most studies were published in English ($n = 381$, 96·7%), some were also written in Mandarin ($n = 7$, 1·8%), Turkish ($n = 2$, 0·5%), Portuguese ($n = 2$, 0·5%), Persian ($n = 1$, 0·3%), and Korean ($n = 1$, 0·3%). Most studies addressed the effect of medications in animal models ($n = 377$, 96%). Seventeen (4%) studies, reporting 22 experiments (5%), reported results in humans. 774 drugs are known to be administered in the acute phase of SCI[7]. 116 (15%) of those drugs were included in experiments identified in our review. 110 drugs were examined individually and 33 in combination (Supplementary Data S1). Six drugs were only tested as part of combinatorial treatments - aminocaproic acid[24], rosuvastatin[25], magnesium chloride[26, 27], ketamine[28], isoflurane[28] and nitroprusside[29].

## Pre-clinical studies

*Population studied* Rat models were most extensively investigated ($n = 382/464$ experiments, 82%). Larger mammals (i.e., cats, dogs) were mainly used before 2001 ($n = 19/22$ experiments conducted on cats and dogs, 86%, Supplementary Fig. S1B). By contrast, all experiments performed on mice ($n = 38$) were published after 2000. Sample size, age, and sex were partly or fully missing in 77 (17%), 341 (73%), and 61 (13%) experiments, respectively. Partly missing entries included sample size bounded or expressed as ranges, age described as adult or young, and samples comprising both male and female in unknown proportions. Likewise, exclusion or death of animals was only reported for 51 (11%) experiments. Among experiments reporting sample size, cohorts included a mean of 63 animals (SD: 52, median: 48, Q1-Q3: 32-80). Studies using larger mammals exhibited smaller cohorts (Supplementary Table S3). When reported, age was commonly expressed in weeks ($n = 87$, 19%). Rats had a mean age of 10 weeks (10.69 weeks when mean age is reported [$n = 31/77$], 8.92 weeks for lower bound and 10.76 weeks for upper bound when ranges are reported [$n = 57/77$]). Mice were also 10 weeks of age (mean of 10.00 [$n = 2/10$], 8.25 [$n = 8/10$] and 10.13 [$n = 8/10$] weeks when mean, minimum and maximum are reported, respectively). A majority of studies included exclusively male or female animals ($n = 387$, 83%), with more experiments being performed on exclusively male populations ($n = 206$, 44%). Details on the use of male, female and mixed populations over time are reported in Supplementary Fig. S1C.

*Injury characteristics* SCI models have been previously categorized into contusion, compression, distraction, dislocation, transection and chemical models[30]. 278 (60%), 132 (28%), 27 (6%), 16 (3%), 7 (2%), 5 (1%) experiments reported a contusion, compression, transection, ischemia, multiple or other injury mechanisms (photochemical lesion[31–33], irradiation[34], electrolytic lesion[35]), respectively. Although

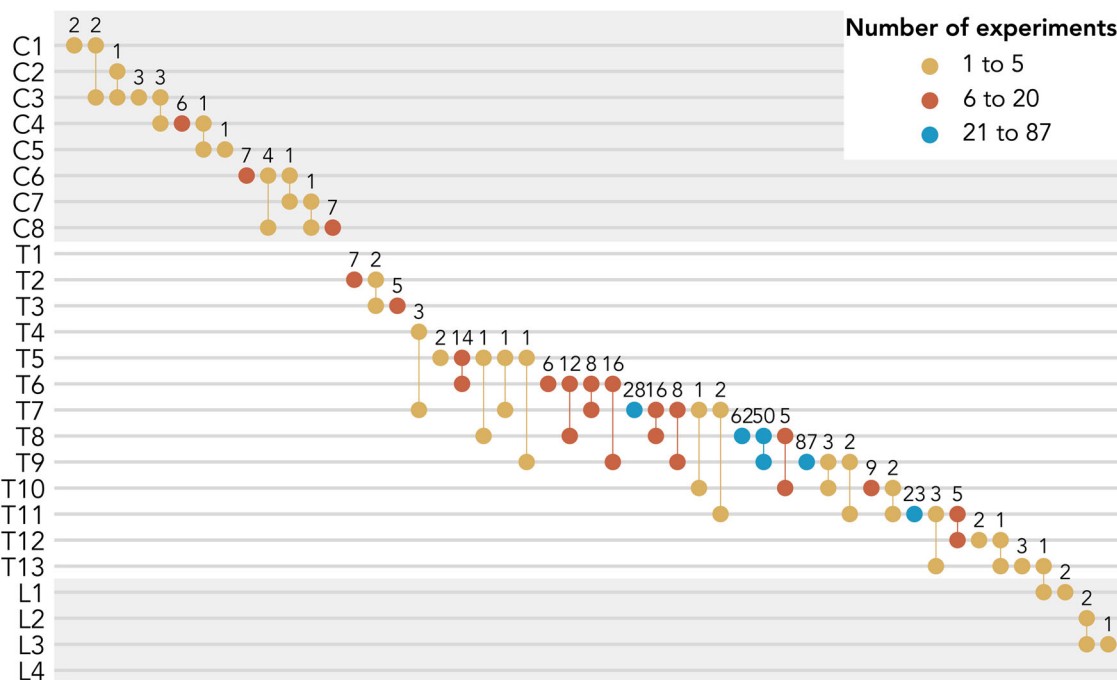

**Fig. 2 | Number of experiments per level of injury studied in animal models of spinal cord injury.** Shaded areas distinguish between cervical, thoracic, and lumbar injuries. Notably, thoracic injuries were the most prevalent in animal experiments.

protocols used to induce injuries were often described in detail, information about the corresponding severity of the injury was missing for most experiments ($n = 257$, 55%). Level of injury was typically reported either precisely ($n = 262$, 56%) or in ranges ($n = 172$, 37%). Most experiments studied injuries at the thoracic level, predominantly at or below T5 ($n = 222$, 85% and n = 151, 88%, of experiments reporting unique and range levels respectively, Fig. 2).

*Drugs investigated and assessments of their effects* 109 individual drugs and 32 combinations were tested in SCI animal models. Methylprednisolone and methylprednisolone sodium succinate were most prevalent among experiments reported with 71 (15%) and 23 (5%) experiments, respectively (Fig. 3). A total of 60 (43%) unique drugs or combinations were tested in more than one experiment.

Drug effects were evaluated by a wide range of neurological and locomotor assessments. The most common choice was the Basso Beattie Bresnahan[36] (BBB) scale, developed and employed for rats. Its original or modified versions (e.g., Basso mouse scale[37], canine BBB locomotor scale[38]) were used in 275 (59%) of the experiments (Fig. 4). Overall, most tests performed ($n = 620/848$, 73%) evaluated locomotor function. One experiment or study could include more than one assessment, and 174 (46%) unique studies tested more than one category among locomotion, skilled forelimb function, sensory function, electrophysiology, and other functional assessments. While assessment protocols were mostly well described, timing, number of repeats, and follow-up period varied widely between experiments.

Figure 3 illustrates the drug effect reported for the most prevalent drugs in our review. One can notice that diverging findings were reported when testing the same drug in different experiments. Using methylprednisolone as an example, 31 experiments reported positive effects, while 28 experiments found no effect for methylprednisolone. Similarly, metformin, atorvastatin, lithium, valproic acid, melatonin, and estradiol were investigated in more than five independent experiments and the majority (>50%) of those experiments reported a positive effect of the treatment (80%, 78%, 63%, 60%, 57%, 56%, respectively). Interestingly, we identified two drugs with negative effects reported (morphine[39], ethanol[28]). However, most of the experiments published and reviewed here found their respective drugs of interest to have a positive ($n = 195$, 42%) or no effect

($n = 115$, 25%) on neurological or functional recovery following SCI. Details of mixed effects reported are presented in Supplementary Fig. S2. A summary of compounds identified for further investigations is provided in Table 2.

## Clinical studies

We extracted information from 17 studies reporting 22 experiments conducted on human cohorts with SCIs (Figs. 5, 6). Cohort sizes varied greatly ($n = 10$[40] to $n > 2000$[41]). Sex distributions were consistently skewed towards male population (from 53.4% to 100% male), in line with the sex distribution observed in the general SCI population[42–45]. While one study (two experiments) explicitly included pediatric participants[46], most experiments considered only adult participants with mean age between 32.5[4] and 57.6[47] years, matching the age distribution reported in the literature[44, 45, 48, 49].

As expected and in contrast to animal studies, most human experiments were performed on heterogeneous groups with regards to their injury characteristics (neurological level of injury, severity, mechanism of injury). The majority of the studies ($n = 15$, 18 experiments) investigated patients with acute SCI. Only three studies (five experiments) specifically enrolled participants with subacute or chronic incomplete injuries[40, 50, 51] comparing test performances pre- and post-exposure to the drugs of interest.

Drugs tested included naloxone[46, 52, 53] ($n = 3$, 14%), cyproheptadine[40, 50] ($n = 2$, 9%), escitalopram[40, 50] ($n = 2$, 9%), baclofen[4], minocycline[54], levodopa[51], testosterone[55] and a combination of progesterone and vitamin D[56] ($n = 1$, each, 5%). Methylprednisolone was the most studied drug ($n = 10$, 45%) with publications between 1990[46] and 2018[57].

All studies evaluated drug effects through neurological assessments. Additionally, functional outcomes such as mobility[40, 50, 58] or spasticity[40, 50] were tested in eight experiments, and one study (two experiments)[40] reported electrophysiological outcomes. Lastly, recovery was assessed based on changes to injury severity in four experiments[4, 47, 57, 59].

Results reported for the effects of methylprednisolone diverged from the animal studies with only one experiment recording positive results[46], which was part of the oldest study of methylprednisolone in humans. Most of the experiments on methylprednisolone reported no effect ($n = 6$, 60%) and three observed mixed effects depending on subgroup[60], assessment[57] or timing of treatment[52]. A similar trend was observed when considering all

**Fig. 3 | Effects on outcomes after spinal cord injury reported for drugs studied in at least five animal experiments.** Circle size is proportional to the number of experiments reporting the effect of interest. Circles are colored proportionally to the frequency that the effect of interest represents among all experiments studying the drug of interest.

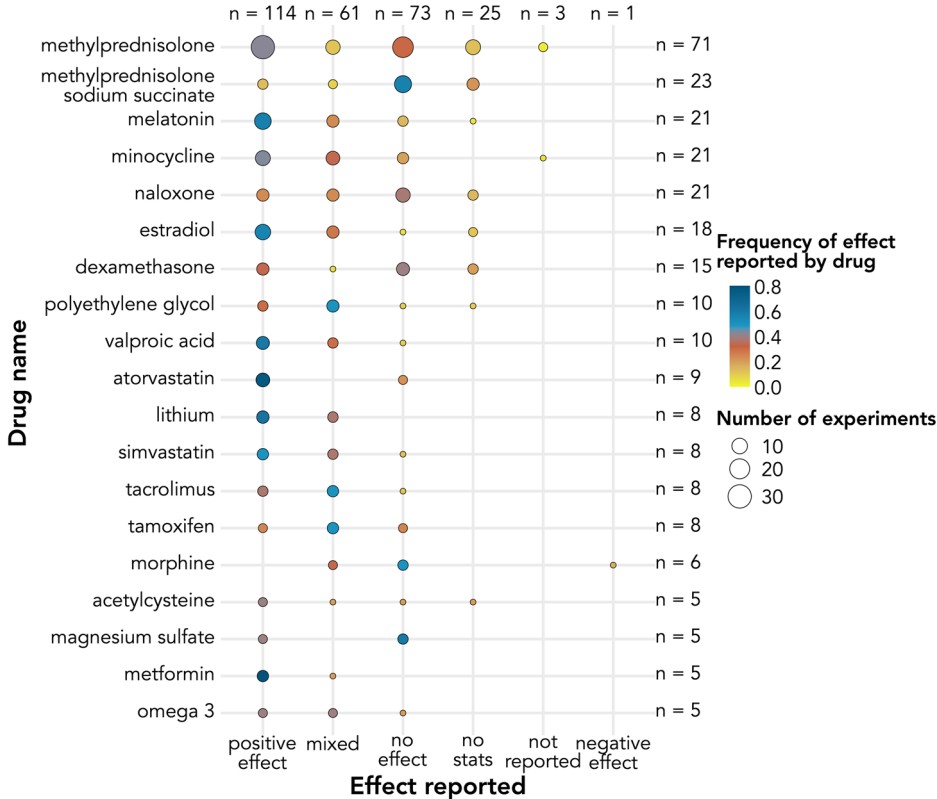

**Fig. 4 | Number of experiments per assessment reported in animal studies.** Individual assessments are grouped into locomotion, skilled forelimb function, sensory function, electrophysiology (EP) and other functional assessments. BBB Basso, Beattie, and Bresnahan locomotor scale, BMS Basso mouse scale, SEPs somatosensory evoked potentials, MEPs motor evoked potentials, SCEPs spinal cord evoked potentials, EP electrophysiology.

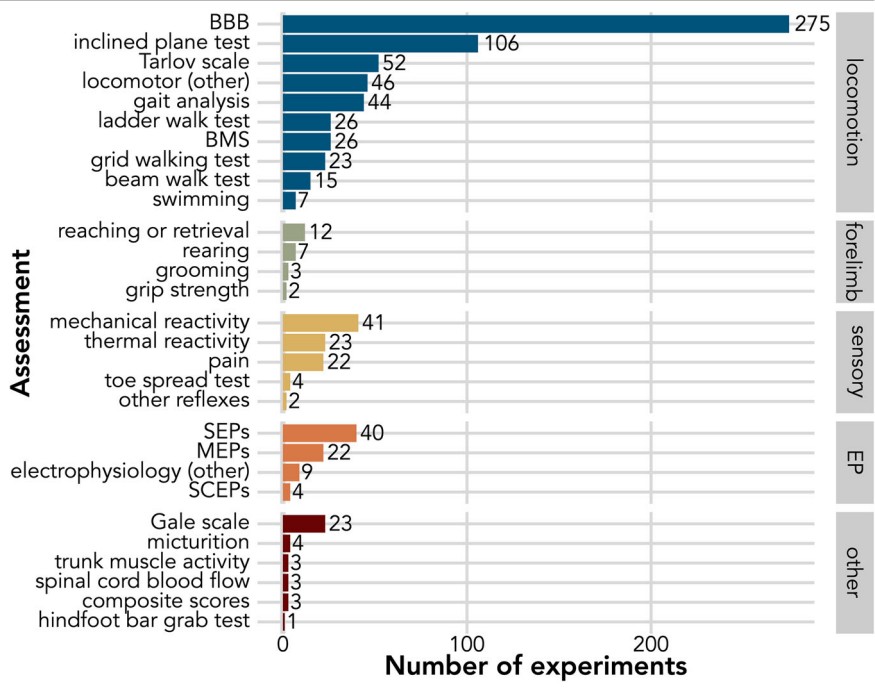

drugs tested in human populations: a total of 12 experiments reported no effect (55%) and 9 described mixed results (41%), mainly due to differences between assessments ($n = 7$, 32%). Notably, most of the data from human populations were collected prospectively ($n = 18/22$, 82%), i.e., individuals were followed and data were collected over time, while they were most often analyzed retrospectively ($n = 12/22$, 55%), i.e., data analysis was planned and completed after the final outcome was known. This hints towards few clinical trials testing pharmacological treatments for SCI.

## Risk of bias (RoB)

RoB was assessed for animal, RCTs, and observational human studies separately. Overall, the majority of animal studies presented with unclear RoB, due to limited reporting on the items targeted by the SYRCLE tool. In

**Table 2 | Summary of compounds identified for further investigations (both in beneficial and detrimental effects) in this review**

| Drug name | Proposed mechanism(s) of action | Results reported in this review | Further references |
|---|---|---|---|
| metformin | Activation of adenosine monophosphate-activated protein kinase signaling and reduction of reactive oxygen species production[93] | 4 experiments with positive effects reported<br>1 experiment with mixed effects (stats/no stats) reported | 94 |
| atorvastatin | Anti-inflammatory[95], neuroprotective effects by reducing the levels of inflammatory factors such as tumor necrosis factor α (TNF-α), and interleukin 1β (IL-1β)[96] | 7 experiments with positive effects reported<br>2 experiments with no effect reported | 97 |
| lithium | Anti-inflammatory[98], inhibition of glycogen synthase kinase-3 beta activity leading to the promoted axonal growth and reduced neurotoxin-induced cell death[99] | 5 experiments with positive effects reported<br>3 experiments with mixed effects (stats/no stats, or dosage, or assessments) reported | 100 |
| valproic acid | Inhibition of histone deacetylases, involved in the regulation of the expression of inflammatory genes[101, 102] | 6 experiments with positive effects reported<br>3 experiments with mixed effects (stats/no stats, or dosage, or a combination of assessments and stats/no stats) reported<br>1 experiment with no effect reported | 103 |
| melatonin | Anti-inflammatory, apoptosis inhibition, reduction of oxidative stress via the regulation of malondialdehyde, glutathione, superoxide dismutase, and myeloperoxidase, regulation of nitric oxide synthase[104] | 12 experiments with positive effects reported<br>5 experiments with mixed effects (stats/no stats, or assessments, or a combination of assessments and regime) reported<br>3 experiment with no effect reported<br>1 experiment with no statistics reported | 104 |
| estradiol | Apoptosis inhibition, improved axon integrity and sparing, reduced myelin degradation, protection against reactive oxygen species[105] | 10 experiments with positive effects reported<br>5 experiments with mixed effects (assessments, or dosage, or a combination of assessments and dosage) reported<br>1 experiment with no effect reported<br>2 experiment with no statistics reported | 105, 106 |
| morphine | No proposed mechanism of action on neurologic or functional recovery found | 1 experiment with negative effects reported<br>2 experiments with mixed effects (assessments) reported<br>3 experiments with no effect reported | 107 |
| ethanol | Pathological radical reactions, increased hemorrhage[108], increased tissue magnesium depletion, increased total tissue lactate levels, more generally worsening of secondary damage[28] | 4 experiments (3 studies) with negative effects reported (note: 2 experiments [1 study] tested ethanol in combination with other drugs [isoflurane; ketamine and pentobarbital]) | 109 |

particular, items corresponding to selection (sequence generation and allocation concealment), performance (random housing and blinding), and attrition (incomplete outcome data) biases were rated as unclear for 59.2%, 92.2%, 99.6%, 91.8%, and 87.5% of the experiments included, respectively. An important other source of bias identified was the frequent use of additional drugs, including anesthetics, painkillers and antibiotics. The grading of incomplete reporting of basic information relating to the study protocol scored from zero to 12, with 36 experiments (7.8%) having a score greater or equal to six (Supplementary Data S4). Variables most affected by incomplete reporting were age and blinding of recovery assessments (Fig. 7a, b). Among observational human studies, only one showed critical RoB (Fig. 7c), while most RCTs showed high RoB in the selection of the reported results (Fig. 7d).

## Discussion

The current study aimed to systematically review existing literature assessing effects of drugs commonly administered in the acute phase of SCI. Encouragingly, several drugs have been investigated across multiple animal models and have consistently demonstrated positive effects[10, 61–64]. This convergence of evidence prompted the formulation of drug repositioning, also known as drug repurposing, as a novel translational approach in the field of acute SCI care. Repositioning has emerged as a successful strategy in other fields (e.g., amantadine in Parkinson's disease[65] and l intuzumab in Alzheimer's disease[66]) to improve neurological outcomes in the absence of novel therapies. Drug repositioning aims at identifying new uses for approved or investigational drugs that are outside the scope of the original drug indication[67]. A clear advantage of this approach is the use of de-risked compounds with established safety and biological activity profiles, thereby reducing overall development costs and shortening timelines[68, 69]. While drug repositioning utilizes existing evidence to accelerate the development of new treatments, it is still affected by challenges of translational research.

We identified 377 studies considering the effects of drugs previously identified as administered to human patients with acute SCI. Evidence exists for 112 (77.8%) unique compounds or combinations to exert beneficial and/or detrimental effects. For example, metformin is routinely used in humans to manage high blood sugar levels caused by type 2 diabetes[70]. Preclinical studies have identified enhanced regeneration in the spinal cord related to metformin-induced autophagy via the mTOR signaling pathway[10, 71, 72]. These observations suggest that administering metformin early after injury could potentially improve long-term neurological outcomes.

Detrimental effects were also observed for some drugs, including opioids, which attenuated the recovery of locomotor function and exacerbated pathophysiological processes in rodent models of SCI[73–76]. A detrimental opioid effect is in line with beneficial effects of naloxone, an opioid antagonist[77, 78], and highly concerning in light of the ubiquitous administration of opioids for pain management in the early stages of SCI. Completely removing or restricting opioids presents serious ethical concerns (i.e., weighing the management of acute pain with long-term neurological effects). However, minimizing the administration of opioids could potentially facilitate neurological recovery[68, 69].

To allow for a comprehensive characterization of potential effects of commonly administered drugs on neurological recovery, we deliberately decided to include preclinical studies involving animal models and clinical studies in humans. Nonetheless, the high degree of heterogeneity across studies, even in a single species, was surprising. A large variability in population characteristics, exact administration parameters, and timing of assessment is observed. In combination with a wide range of spinal levels subjected to injury and different species being studied, comparisons between experiments are challenging or impossible. One exception is the study by Popovich et al.[79] aiming to replicate findings, which noted a strong connection between initial injury characteristics and detectable drug effects. This highlights the need for varying as few parameters as possible to allow for meaningful comparisons. In human studies we suspect that the majority reporting mixed or no effects also reflects the heterogeneity in injury

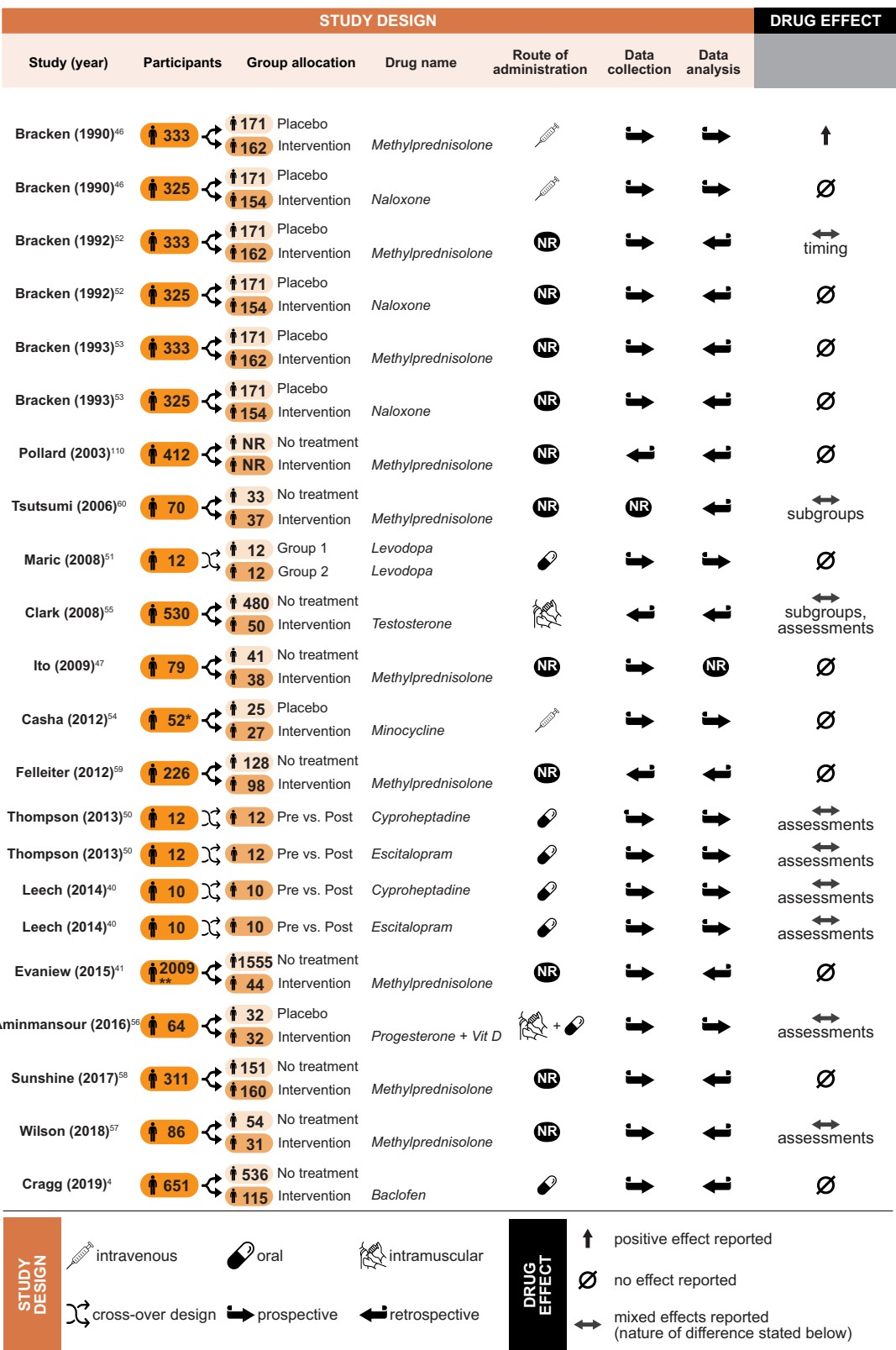

**Fig. 5 | Overview of study design and outcome for studies of spinal cord injury in humans.** Icons are used to indicate the route of administration (intravenous, oral, intramuscular) of the drug of interest, and central aspects of study design, where prospective and retrospective refer to how data collection and analysis were planned and completed, while cross-over refers to a study design in which measurements taken pre- and post-exposure to a drug of interest were compared, which deviates from a split of individuals into control and treatment otherwise used. *: 71 participants were recruited for this study, of which 19 did not meet the inclusion criteria;

**: 410 patients were excluded from the analysis in this study, making up the difference between sizes reported for the no treatment and intervention groups. Vit D: Vitamin D. The figure includes the following icons obtained from thenounproject.com: "Direction" icon by Uswa KDT CC BY 3.0, "Pill" icon by Three Six Five CC BY 3.0, "Shuffle" icon by Gregor Cresnar CC BY 3.0, "Syringe" icon by arman maulana CC BY 3.0, "Vaccination" icon by WiStudio CC BY 3.0, and "Empty set" icon by Gregor Cresnar CC BY 3.0.

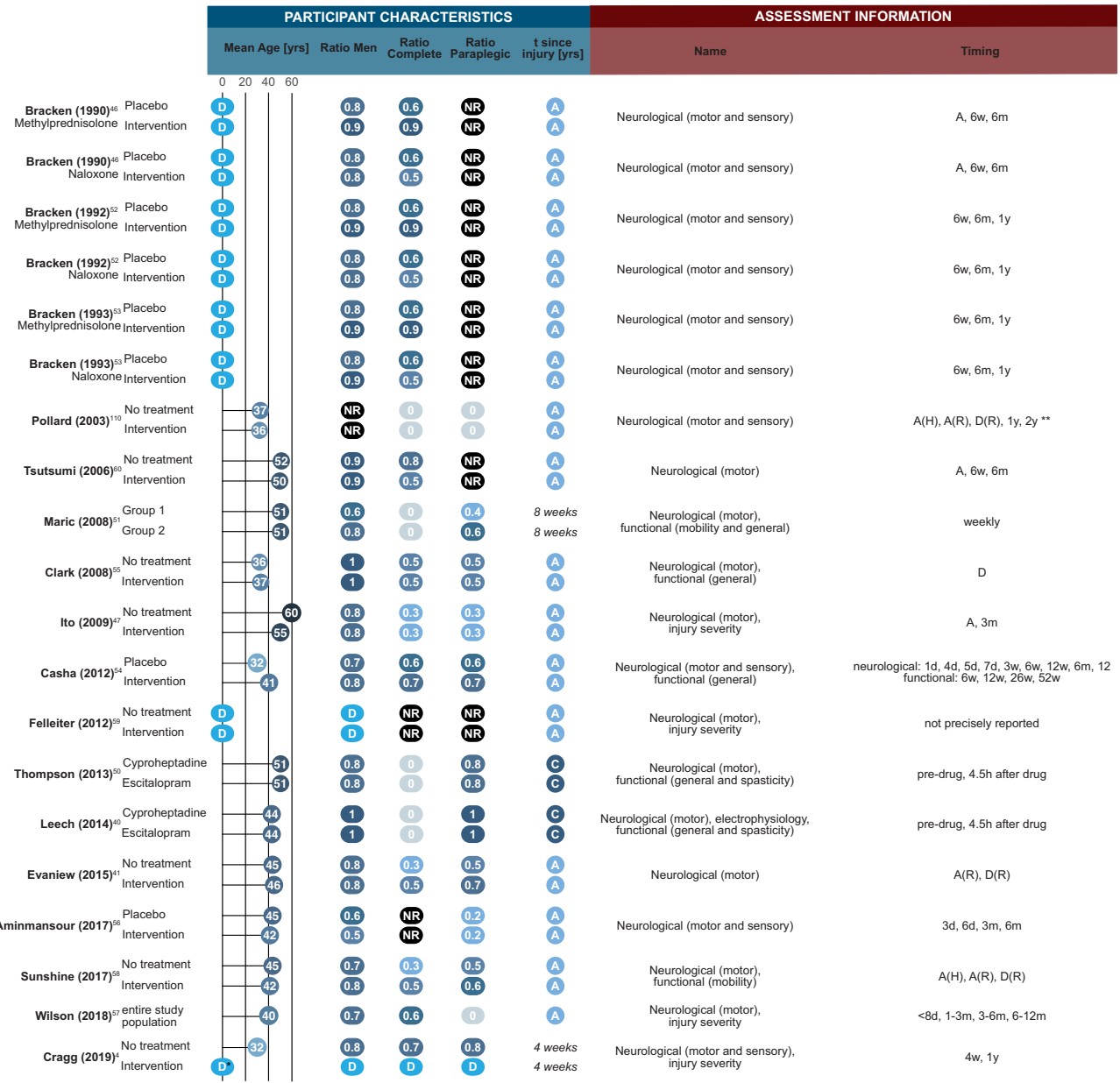

**Fig. 6 | Overview of participant characteristics and assessment details for studies of spinal cord injury in humans.** D only distributional information reported, NR information not reported, A participants recruited in acute phase after injury, C participants recruited in chronic phase after injury, h: hours, d days, m months, w weeks, y: year, yrs: years, A(H) acute phase hospital stay, A(R) acute phase rehabilitation stay, D(R) discharge from rehabilitation, *: Age (mean and standard deviation) reported for low and high dose groups separately; **: 1 y and 2 y reported if possible but not mandatory.

patterns included. This likely results in effects which vary widely between individuals and cannot be detected in a group-level analysis. This extensive heterogeneity means that, currently, meta-analyses are not feasible, even for the most commonly studied drugs (Fig. 8). This constitutes a notable limitation as the large fraction of positive effects reported might hint towards a publication bias (Section "Risk of bias (RoB)"). Methylprednisolone constitutes the most interesting example of this pattern, as it has been extensively studied in both pre-clinical and clinical environments with mostly positive results or no effect reported (Figs. 3, 5). While methylprednisolone is still used as an active control in some animal studies, it is no longer an accepted treatment for acute SCI in humans.

The lack of an effective pharmacological treatment for SCI highlights the discrepancy between largely positive pre-clinical results and unsuccessful translation to human subjects. A number of hypotheses that could explain this divergence can be derived from this review. One noticeable difference concerns basic study parameters such as the age of the cohort studied or level of injury. While the age distribution in humans affected by SCI is moving towards a bimodal shape[45], studies in animal models are typically performed on more homogeneous groups of younger individuals[80–84]. While the use of young animals might be a result of practical limitations or cost reduction, the mean age at time of injury (10.0 weeks) can be approximately projected to an 18 year old human[85], which fails to capture the human population of SCI and potentially affects translatability. Further, SCI in humans occurs predominantly in the cervical segment of the spinal cord[45], while animals are mostly injured in the thoracic region (Fig. 2), likely due to ethical requirements. Similarly, injury severity has been named as a critical parameter to control for in animal studies to ensure translatability of findings to the human population[86]. Unfortunately, we found that it was also one of the factors least frequently reported (45% of studies). Noticeable differences also exist in the administration of drugs. Animal studies typically

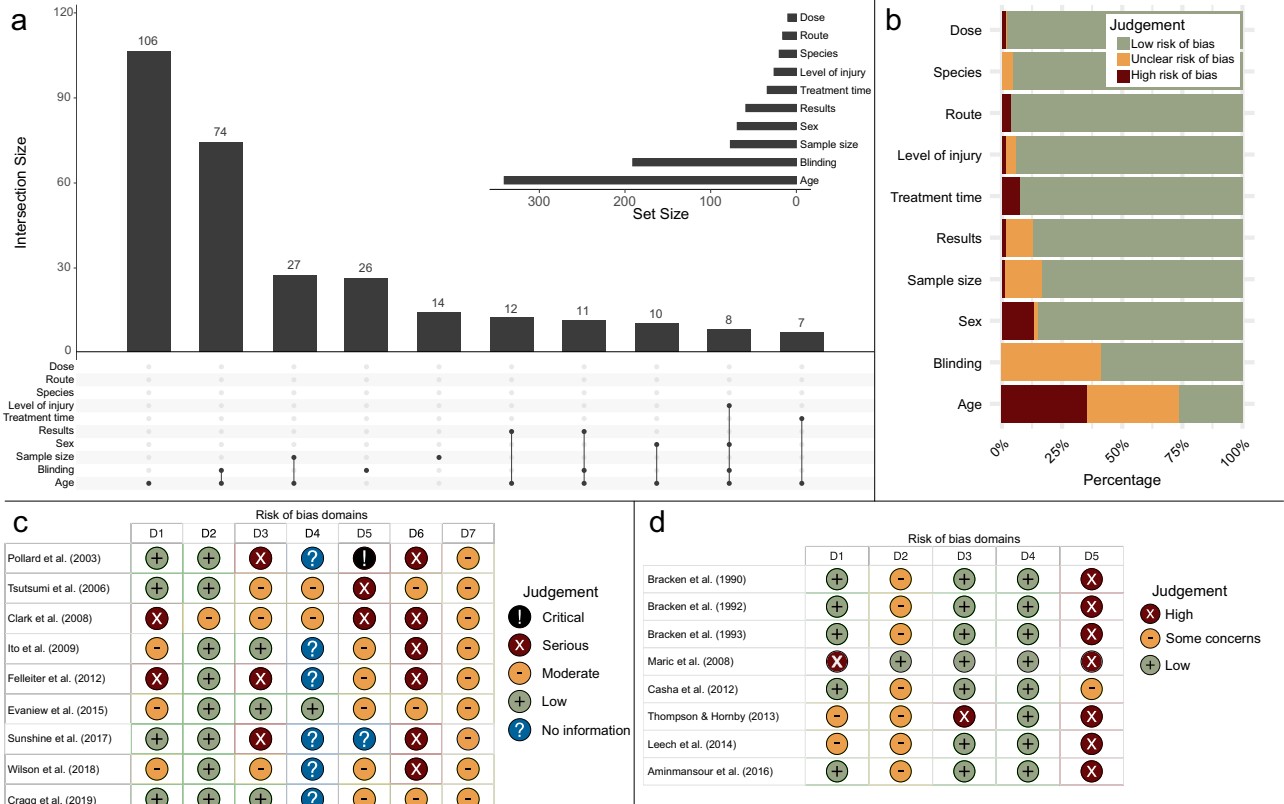

**Fig. 7 | Assessment of the risk of bias for included experiments (animal and human studies). a** Co-occurrence of potential bias (grading as low or high risk) within animal experiments. Risk of bias was most prevalent in reported age, followed by a combination of age and blinding status. Conversely, information on species, route of drug administration, and dose showed lower risk of bias. **b** Proportion of each risk of bias (low, unclear or high) by domain of bias studied. Age represents the domain with the most prevalent high risk of bias. **c** Risk of bias for human intervention studies (observational). Only one study showed a critical risk of bias (domain 5: bias due to missing data), whilst most studies ($n = 6, 67\%$) did not provide sufficient information to assess the risk of bias due to deviations from intended

interventions (domain 4). Additionally, the majority of the studies ($n = 8, 89\%$) had a low risk of bias due to selection of participants (domain 2). D1 bias due to confounding, D2 bias due to selection of participants, D3 bias in classification of interventions, D4 bias due do deviations from intended interventions, D5 bias due to missing data, D6 bias in measurement of outcomes, D7 bias in selection of the reported result. **d** Risk of bias for randomized controlled trials (RCTs). High risk of bias was detected in 7 studies (88%) for bias in selection of the reported results. D1 bias arising from the randomization process, D2 bias due to deviations from intended interventions, D3 bias due to missing outcome data, D4 bias in measurement of outcome, D5 bias in selection of reported result.

follow a weight-based dosing regime while humans receive a standardized dose. Similarly, many animal studies initiate treatment immediately after injury[64, 87, 88], which appears infeasible in the human population. Additionally, most pre-clinical studies would restrict their investigation to a single drug, while human populations are subject to a large polypharmacy, with up to 59 drugs prescribed in the acute phase[7]. Translatability of findings from preclinical studies might be hindered as most of the preclinical studies fail to account for interactions between the drug under investigation and other compounds (e.g., treatments for pain management or other complications). These issues in the transfer from animal to human studies might contribute to the majority of human studies reporting mixed effects. While beneficial effects might still exist in humans, they could go undetected due to the scarcity of RCTs (Section "Clinical Studies"). While RCTs require substantial resources and can be challenging to conduct in a rare and heterogeneous condition like SCI, advancements in the treatment of SCI will only be possible if efforts extend from preclinical studies to systematic prospective data collection and analysis in humans. Finally, only a small subset of studies in humans considers the effect of drugs in the chronic phase. While animal studies often include chronic injury models (for examples see[89–91]) no chronic animal studies were encountered in the scope of this systematic review. One explanation could be that the hypothesized effects of drugs of interest selected target mechanisms of repair which are active early after injury more than at the chronic stage. It would however be interesting

to see more human and animal chronic SCI studies investigating the effects of these drugs on debilitating secondary long-term complications[86].

A noteworthy limitation of the current review was that literature search was limited to articles listed in PubMed/Medline, Scopus, and Web of Science, or identified by hand searches. Considering the pace at which research in this area advances, it is likely that the findings of the publications described in this paper will be quickly complemented by further research. The literature search also excluded gray literature (e.g., preprints, reports, conference proceedings), the importance of which to this topic is unknown, and thus might have introduced another source of search bias. Publication bias is likely to result in studies with positive results being preferentially submitted and accepted for publication.

The present review provides an extensive summary of existing evidence on effects of drugs administered to individuals affected by SCI. In particular, results highlight melatonin, estradiol, and valproic acid as commonly investigated drugs with largely positive effects, indicating the inherent potential to advance treatment through drug repurposing. Simultaneously, we observed and extensively characterized sources of heterogeneity among the valuable resources provided by existing studies. In light of the current lack of an effective pharmacological treatment for SCI and failed attempts to develop new treatments, the field would benefit from further standardization in studying and reporting drug effects investigated in animal models.

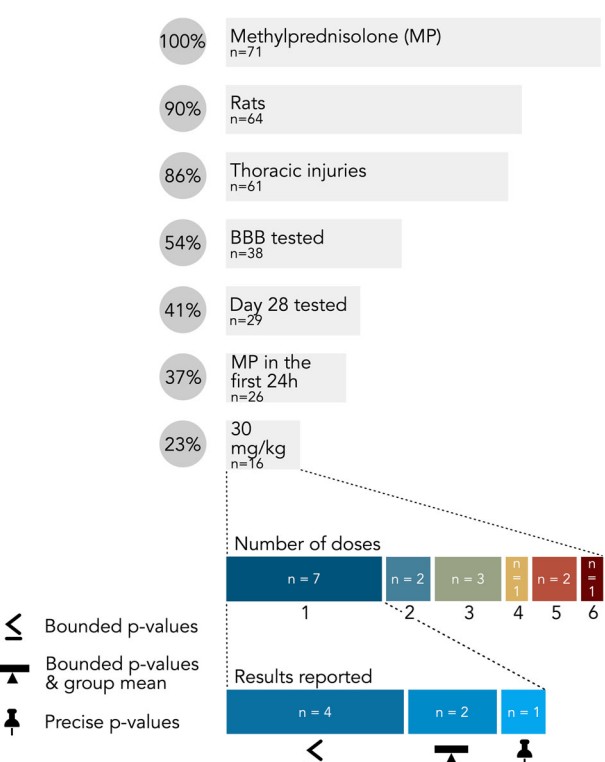

**Fig. 8 | Illustration of the heterogeneity observed among animal experiments reporting effects of methylprednisolone on neurological and functional recovery after SCI.** Numbers reported in the bars refer to the size of the respective subset of studies with the relevant characteristics. The figure includes the following icons obtained from thenounproject.com: "Average" icon by James Bickerton CC BY 3.0, "Less or equal" icon by Julia Holmberg CC BY 3.0, and "Pin" icon by Nice Design CC BY 3.0.

## Data availability

The data used in this study can be accessed at our GitLab repository (https://gitlab.ethz.ch/BMDSlab/publications/SCI-drug-review-publication). Supplementary Data S1 contains information on the number of studies reporting on individual drugs of interest or combinations thereof, which is summarized for the most frequently studied drugs in Fig. 3. Supplementary Data S2 contains information on the assessments used as part of animal studies and the grouping applied, which is presented in Fig. 4. Supplementary Data S4 contains information on the risk of bias assessment applied to animal studies, which is summarized in Fig. 7a, b.

## Code availability

The source code for the analysis performed, including visualizations, can be accessed at our GitLab repository (https://gitlab.ethz.ch/BMDSlab/publications/SCI-drug-review-publication) and Zenodo[92]. R Statistical Software version 4.3.1 and Python version 3.10.10 were used for all analysis.

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

## Acknowledgements

This study was supported by the Swiss National Science Foundation (Ambizione Grant, #PZ00P3_186101 and #IZLIZ3_200275, C.R.J.), Wings for Life Research Foundation (#2017_044, #ID 2020_118, C.R.J. and J.L.K.K.), the International Foundation for Research in Paraplegia (#P192, C.R.J.). L.B. was supported by an ICORD student exchange award provided by the Praxis Spinal Cord Injury Foundation. B.R.K. was supported by the Paralyzed Veterans of America Research Foundation (# 3195). The authors would like to thank Najmeh Kheram for her support in analyzing studies published in Persian. The following icons from thenounproject.com were used: "Average" icon by James Bickerton CC BY 3.0; "Less or equal" icon by Julia Holmberg CC BY 3.0; "Pin" icon by Nice Design CC BY 3.0; "Direction" icon by Uswa KDT CC BY 3.0; "Pill" icon by Three Six Five CC BY 3.0; "Shuffle" icon by Gregor Cresnar CC BY 3.0; "Syringe" icon by arman maulana CC BY 3.0; "Vaccination" icon by WiStudio CC BY 3.0; "Empty set" icon by Gregor Cresnar CC BY 3.0. *Statistical Analyses completed by*: L.B. and L.P.L. (Swiss Federal Institute of Technology, ETH Zurich).

## Author contributions

All authors had full access to all the data in the study and had final responsibility for the decision to submit for publication. L.B.: selection of studies, extraction of data, statistical analysis, visualization, interpretation of the data, drafting of the manuscript; L.P.L.: selection of studies, extraction of data, statistical analysis, visualization, interpretation of the data, drafting of the manuscript; B.R.K.: interpretation of data, revising the manuscript for intellectual content; B.T.: extraction of data, interpretation of data, revising the manuscript for intellectual content; J.J.L.: extraction of data, interpretation of data, revising the manuscript for intellectual content; T.G.: extraction of data, interpretation of data, revising the manuscript for intellectual content; W.T.: interpretation of data, revising the manuscript for intellectual content; J.L.K.K.: interpretation of data, revising the manuscript for intellectual content; M.W.: study design, selection of studies, interpretation of data, revising the manuscript for intellectual content; C.R.J.: study design, selection of studies, extraction of data, visualization, interpretation of the data, drafting of the manuscript.

## Competing interests

M.W. has received/is receiving funding from Michael Smith Foundation for Health Research in partnership with the Rick Hansen Foundation, University Hospital Basel, Wellspect, Coloplast, Stoke Mandeville Spinal Research, International Foundation for Research in Paraplegia, and Swiss Multiple Sclerosis Society. M.W. reports advisory board activity for Coloplast from 2022 to 2024.
