## [Peer Review File · Communications Medicine]

Reviewers' comments:

Reviewer #1 (Remarks to the Author):

This is an excellent, comprehensive review of studies (preclinical and clinical) examining the effects of commonly administered drugs after SCI. As noted, clinical administrations of standard-of-care drugs have been largely overlooked as a disease-modifying factors, but they may have significant implications for recovery. The paper is well-written and clear. As minor suggestions:

1. It would be helpful to add subheadings in the results, to help orient the reader
2. That patients receive up to 60 unique drugs within the first two months, often in combinatorial fashion is staggering. This points also warrants mention in the discussion. Are studies in animal models, that for the most part don't combine treatments, overlooking synergistic/antagonistic effects of drug treatments?

Reviewer #2 (Remarks to the Author):

The authors present an interesting and useful summary of commonly administered drugs and their effects on functional outcomes after spinal cord injury.

Abstract

Please address consistency differences in "disease modifying" or "disease-modifying"

Plain language summary:

I am not convinced that "polypharmacy" is plain language. This summary also seems to include animal experiments in the numbers of drug combinations administered experimentally to "patients" which is misleading.

Introduction:

It is worth also highlighting more clearly the significance of determining drugs which may have a negative effect on neurological outcome after SCI. This is a significant value of your study which is underemphasised in its current form.

Methods

Please expand on how your original paper identified "drugs of interest". Based on your PRISMA flow chart, a lot of drugs have been excluded for this reason

Please clearly state which ROB tools were used.

Results

Figure 3 - the difference in symbol used for indication of prospective vs retrospective is too subtle to clearly see.

Figure 4 - some of the text is too small to discern, even when zoomed in

Discussion

Please include on first use of "repositioning" the alternative terminology (repurposing) for clarity.

Line 436 - deliberately decided

Line 452 "allows to formulate" - revise

Line 457 - translatability rather than transferability

Line 456 - this is probably also a reflection of reduced cost of using younger animals, and a general lack of study of the effect of aging on SCI outcome/therapeutics in pre-clinical studies

Line 457 - the use of thoracic injury is due to ethical limitations

Line 458 - your discussion of the lack of reporting of injury severity in pre-clinical studies is confusing, please expand

Line 470 - there are a number of publications using chronic SCI models

General

Thousands should be separated by commas, not apostrophes.

It is worth elucidating the point that it is generally accepted that methylprednisolone is not beneficial in SCI, despite the conflicting results your review has identified.

To improve the impact of this work, I also suggest inclusion of a table or similar to identify the most promising agents/combinations identified in your review, perhaps the top ten, with some short summary or reference to their proposed mechanism of neuroprotective/neurorestorative action. (i.e the most promising drugs from pre-clinical studies which have not yet been robustly disproven in clinical studies) perhaps including reference to systematic reviews performed elsewhere (for example valproic acid here: [10.1016/j.wneu.2023.10.135](https://doi.org/10.1016/j.wneu.2023.10.135))

The authors would like to kindly thank both reviewers and the editor for taking the time to read our manuscript and provide feedback to improve it. We have worked on the suggestions made and would like to share the resulting updated manuscript.

Reviewer #1 (Remarks to the Author):

This is an excellent, comprehensive review of studies (preclinical and clinical) examining the effects of commonly administered drugs after SCI. As noted, clinical administrations of standard-of-care drugs have been largely overlooked as a disease-modifying factors, but they may have significant implications for recovery. The paper is well-written and clear. As minor suggestions:

1. It would be helpful to add subheadings in the results, to help orient the reader

Thank you for your suggestions, we have added the following subheadings:

3.1 Preclinical studies

3.1.1 Population studied

3.1.2 Injury characteristics

3.1.3 Drugs investigated and assessment of their effects

3.2 Clinical studies

3.3 Risk of bias

2. That patients receive up to 60 unique drugs within the first two months, often in combinatorial fashion is staggering. This points also warrants mention in the discussion. Are studies in animal models, that for the most part don't combine treatments, overlooking synergistic/antagonistic effects of drug treatments?

Thank you for the suggestion; this point is now included in the discussion in the fourth paragraph on differences between pre-clinical and clinical studied, and addressed as follows: "Additionally, most pre-clinical studies would restrict their investigation to a single drug, while human populations are subject to a large polypharmacy, with up to 59 drugs prescribed in the acute phase ¹⁰⁶. Translatability of findings from preclinical studies might be hindered as most of the preclinical studies fail to account for interactions between the drug under investigation and other compounds (e.g., treatments for pain management or other complications)."

Reviewer #2 (Remarks to the Author):

The authors present an interesting and useful summary of commonly administered drugs and their effects on functional outcomes after spinal cord injury.

Abstract

Please address consistency differences in "disease modifying" or "disease-modifying"

Thank you for noticing, we have harmonized all entries to "disease-modifying".

Plain language summary:

I am not convinced that "polypharmacy" is plain language. This summary also seems to include animal experiments in the numbers of drug combinations administered experimentally to "patients" which is misleading.

Thank you for the suggestion to modify the terminology for better understanding and consistency. We replaced "polypharmacy" with a description of the phenomenon in simpler language: "The effect of *providing patients with a large number of medications in the acute phase after injury* on recovery from SCI, however, is typically not considered." Additionally, we expanded the statement on the number of drugs and drug combinations to indicate that this number derives from animal and human studies: "144 unique drugs or combinations of drugs previously reported to be administered *in animal models or to patients with SCI* have been studied [...]"

Introduction:

It is worth also highlighting more clearly the significance of determining drugs which may have a negative effect on neurological outcome after SCI. This is a significant value of your study which is underemphasised in its current form.

We expanded the second paragraph of the introduction to explicitly highlight this issue, and how our review could improve treatment of SCI in this aspect. The second sentence now reads: "Consequently, understanding the potential therapeutic benefits *or possible harm* of routinely administered drugs on neurological and functional recovery [...]" The following sentence has been added to the paragraph: "Simultaneously, potential harmful effects of commonly administered drugs on neurological recovery are rarely considered but their identification could allow for crucial changes in treatment strategies."

Methods

Please expand on how your original paper identified "drugs of interest". Based on your PRISMA flow chart, a lot of drugs have been excluded for this reason.

We added the following definition of drug of interest in Section 2.1: "The list of *all drugs administered in the first 60 days after injury* to treat secondary complications *in the Sygen¹⁷ and SCIR rehab¹⁸ cohorts* were extracted from our recent publication.⁷ We will refer to those as "drugs of interest". " We made an additional reference to it in paragraph 2.4: "In particular, out-of-scope studies included publications investigating drugs outside of the drugs of interest as defined in Section 2.1."

Please clearly state which ROB tools were used.

Section 2.6 on data extraction has been adapted to clarify this aspect, and now reads: "*Clinical studies on human populations were assessed for risk of bias (RoB) according to their design, either using the RoB 2 tool for randomized clinical trials (RCTs)²¹ or the ROBINS-I tool for non-randomised interventions.²² Animal experiments were assessed for risk of bias from selective reporting and assigned a score from 0 (no bias) to 20 (highest risk of bias) according to criteria listed in Supplementary Table S5. Visualizations for RoB assessments of RCTs and intervention studies were created using robvis.²³"*

Results

Figure 3 - the difference in symbol used for indication of prospective vs retrospective is too subtle to clearly see.

Thank you for your comment, we modified the icons used both in Figure 3 and its legend. Please note that it led to the following changes in the Acknowledgements/Icons section: “

- ~~“Clock” icon by Made x Made Icons CC BY 3.0~~
- ~~“Past” icon by Made x Made Icons CC BY 3.0~~
- “Direction” icon by Uswa KDT CC BY 3.0”

Figure 4 - some of the text is too small to discern, even when zoomed in
Thank you for your suggestion, we adjusted Figure 4 accordingly.

Discussion

Please include on first use of "repositioning" the alternative terminology (repurposing) for clarity.

Thank you for your suggestion. We have added this information as follows: “This convergence of evidence prompted the formulation of drug repositioning, *also known as drug repurposing*, as a novel translational approach in the field of acute SCI care.”

Line 436 - deliberately decided

Thank you, we have made the change as suggested.

Line 452 "allows to formulate" - revise

Thank you for your suggestion, we have revised the sentence as follows: “A number of hypotheses that could explain this divergence can be derived from this review.”

Line 457 - translatability rather than transferability

Thank you, we have made the change as suggested.

Line 456 - this is probably also a reflection of reduced cost of using younger animals, and a general lack of study of the effect of aging on SCI outcome/therapeutics in pre-clinical studies

Thank you for your comment, we added a mention to it in the corresponding sentence: “While the use of young animals might be a result of ethical guidelines *or cost reduction* [...]”.

Line 457 - the use of thoracic injury is due to ethical limitations

Thank you, we added a matching remark to the relevant sentence: “Further, SCI in humans occurs predominantly in the cervical segment of the spinal cord, 43 while animals are mostly injured in the thoracic region (Figure 2A), *likely due to ethical requirements*.”

Line 458 - your discussion of the lack of reporting of injury severity in pre-clinical studies is confusing, please expand

Thank you for your comment, we modified the sentence's structure to make it clearer and expanded our statement as follows: "Similarly, injury severity has been named as a critical parameter to control for in animal studies to ensure translatability of findings to the human population.⁸⁶ Unfortunately, we found that it was also one of the factors least frequently reported (45% of studies)."

Line 470 - there are a number of publications using chronic SCI models

Thank you for your comment on chronic SCI models. We could not identify studies on chronic SCI in animal models in a subsequent literature search, and would be grateful for any pointers to relevant literature. We still adapted the sentence addressing the need for further studies on chronic SCI as follows: "As chronic injuries *are rarely* investigated in animal studies due to ethical restrictions,^{107,108,109} studies of chronic human SCI populations should be *further* expanded to address debilitating secondary complications.¹⁰³"

General

Thousands should be separated by commas, not apostrophes.

Thank you for your suggestion, we adapted the text and figures accordingly.

It is worth elucidating the point that it is generally accepted that methylprednisolone is not beneficial in SCI, despite the conflicting results your review has identified.

Thank you for the suggestion to specifically highlight Methylprednisolone. We included the following sentence towards the end of the third paragraph in the Discussion: "*Methylprednisolone constitutes the most interesting case example of this pattern, as it has been extensively studied in both pre-clinical and clinical environments with positive and negative results reported but its use is generally thought to not be of benefit to recovery from SCI.*"

To improve the impact of this work, I also suggest inclusion of a table or similar to identify the most promising agents/combinations identified in your review, perhaps the top ten, with some short summary or reference to their proposed mechanism of neuroprotective/neurorestorative action. (i.e the most promising drugs from pre-clinical studies which have not yet been robustly disproven in clinical studies) perhaps including reference to systematic reviews performed elsewhere (for example valproic acid here: t10.1016/j.wneu.2023.10.135)

Thank you for the suggestion. We included the suggested information, and pointers to additional literature in a separate element (Box 2) in the manuscript. We refer to this new Box in Section 3.1.3, where we added: "A summary of compounds identified for further investigations is provided in Box 2."

Reviewers' comments:

Reviewer #1 (Remarks to the Author):

The authors have addressed all comments.

Reviewer #2 (Remarks to the Author):

Thank you for the amendments and I am generally satisfied that they have addressed the points raised during the first round of review. I have three points which are outstanding and were not sufficiently addressed in the revision:

Regarding your question about chronic SCI models, a pubmed search found a number of papers on the first page alone that use chronic time points post-SCI. 10.1101/2024.01.10.575021 and 10.1016/j.expneurol.2021.113672 and 10.1016/j.brainres.2023.148484 and 10.1080/01616412.2022.2112380 and 10.1038/s41392-022-01010-1 and 10.1016/j.heliyon.2024.e28522. Perhaps this is not what you mean by chronic SCI models? Either way I think you still need to expand your point here. I'm not suggesting you need to include these references, but they are just an example that there is a lot of literature from animal studies in chronic SCI, beyond the three which you cite. This quite broad literature base rather undermines the point you make.

I suggest that the main reason is that the mechanisms of repair after consolidation are vastly different from attempts at intervention during the acute phase of SCI. For example, attempts at abortion of the glial scar is very different after it is consolidated, and anti-apoptosis treatments are not beneficial after apoptosis has occurred. But chronic models are quite commonly investigated, particularly in relation to neuropathic pain.

Regarding methylprednisolone, it should be more strongly stated that this is not an accepted therapeutic option for SCI (as summarised here 10.3390/biomedicines12030643). The original publication of the National Acute Spinal Cord Injury Study (NASCIS2) in 1990 resulted in widespread implementation of methylprednisolone therapy in SCI on the basis of unclear and inconsistent results, and its inclusion in clinical guidelines has been conflicting in the period since [144,145]. More recent attempts to validate any beneficial effects of SCI have not provided conclusive evidence (Geisler, F.H.; Moghaddamjou, A.; Wilson, J.R.F.; Fehlings, M.G. Methylprednisolone in Acute Traumatic Spinal Cord Injury: Case-Matched Outcomes from the NASCIS2 and Sygen Historical Spinal Cord Injury Studies with Contemporary Statistical Analysis. J.

Neurosurg. Spine 2023, 38, 595–606. Evaniew, N.; Belley-Côté, E.P.; Fallah, N.; Noonan, V.K.; Rivers, C.S.; Dvorak, M.F. Methylprednisolone for the Treatment of Patients with Acute Spinal Cord Injuries: A Systematic Review and Meta-Analysis. *J. Neurotrauma* 2016, 33, 468. Liu, Z.; Yang, Y.; He, L.; Pang, M.; Luo, C.; Liu, B.; Rong, L. High-Dose Methylprednisolone for Acute Traumatic Spinal Cord Injury: A Meta-Analysis. *Neurology* 2019, 93, E841–E850.). Administration of methylprednisolone in TBI has been demonstrated in the CRASH trial to increase the risk of two-week mortality.

Regarding your RoB, the tool in S5 does not appear suitable, as it only has an outcome of unclear or high risk of bias, and does not describe how a score of 20 could be derived. Why have you not used SYRCLE RoB tool, which is the accepted tool for assessment of bias in animal studies. I suggest that you revise the RoB assessment to use an accepted or appropriate measure.

The authors would like to kindly thank the reviewers and editors again for their time and work to read and review this manuscript. We would like to share with you the updated manuscript based on the additional feedback received.

Reviewer #1 (Remarks to the Author):

The authors have addressed all comments.

Thank you for assessing our work again.

Reviewer #2 (Remarks to the Author):

Thank you for the amendments and I am generally satisfied that they have addressed the points raised during the first round of review.

Thank you for assessing our work again.

I have three points which are outstanding and were not sufficiently addressed in the revision:

Regarding your question about chronic SCI models, a pubmed search found a number of papers on the first page alone that use chronic time points post-SCI. 10.1101/2024.01.10.575021 and 10.1016/j.expneurol.2021.113672 and 10.1016/j.brainres.2023.148484 and 10.1080/01616412.2022.2112380 and 10.1038/s41392-022-01010-1 and 10.1016/j.heliyon.2024.e28522. Perhaps this is not what you mean by chronic SCI models? Either way I think you still need to expand your point here. I'm not suggesting you need to include these references, but they are just an example that there is a lot of literature from animal studies in chronic SCI, beyond the three which you cite. This quite broad literature base rather undermines the point you make.

I suggest that the main reason is that the mechanisms of repair after consolidation are vastly different from attempts at intervention during the acute phase of SCI. For example, attempts at abortion of the glial scar is very different after it is consolidated, and anti-apoptosis treatments are not beneficial after apoptosis has occurred. But chronic models are quite commonly investigated, particularly in relation to neuropathic pain.

Thank you for raising this point. We now highlight that we did not observe chronic SCI models in the scope of this systematic review, despite the existence of chronic SCI animal models. We have therefore made the following modifications in the Discussion section: "While animal studies often include chronic injury models (for examples see ^{108,109,110}) no chronic animal studies were encountered in the scope of this systematic review. One explanation could be that the hypothesized effects of drugs of interest selected target mechanisms of repair which are active early after injury more than at the chronic stage. It would however be interesting to see more human and animal chronic SCI studies investigating the effects of these drugs on debilitating secondary long-term complications. ¹⁰⁴".

Regarding methylprednisolone, it should be more strongly stated that this is not an accepted therapeutic option for SCI (as summarised here 10.3390/biomedicines12030643). The original publication of the National Acute Spinal Cord Injury Study (NASCIS2) in 1990 resulted in widespread implementation of methylprednisolone therapy in SCI on the basis of unclear and

inconsistent results, and its inclusion in clinical guidelines has been conflicting in the period since [144,145]. More recent attempts to validate any beneficial effects of SCI have not provided conclusive evidence (Geisler, F.H.; Moghaddamjou, A.; Wilson, J.R.F.; Fehlings, M.G. Methylprednisolone in Acute Traumatic Spinal Cord Injury: Case-Matched Outcomes from the NASCIS2 and Sygen Historical Spinal Cord Injury Studies with Contemporary Statistical Analysis. *J. Neurosurg. Spine* 2023, 38, 595–606. Evaniew, N.; Belley-Côté, E.P.; Fallah, N.; Noonan, V.K.; Rivers, C.S.; Dvorak, M.F. Methylprednisolone for the Treatment of Patients with Acute Spinal Cord Injuries: A Systematic Review and Meta-Analysis. *J. Neurotrauma* 2016, 33, 468. Liu, Z.; Yang, Y.; He, L.; Pang, M.; Luo, C.; Liu, B.; Rong, L. High-Dose Methylprednisolone for Acute Traumatic Spinal Cord Injury: A Meta-Analysis. *Neurology* 2019, 93, E841–E850.). Administration of methylprednisolone in TBI has been demonstrated in the CRASH trial to increase the risk of two-week mortality.

Thank you for your comment. We have updated our discussion on methylprednisolone and added the following sentence: “While methylprednisolone is still used as an active control in some animal studies it is no longer an accepted treatment for acute SCI in humans.”.

Regarding your RoB, the tool in S5 does not appear suitable, as it only has an outcome of unclear or high risk of bias, and does not describe how a score of 20 could be derived. Why have you not used the SYRCLE RoB tool, which is the accepted tool for assessment of bias in animal studies. I suggest that you revise the RoB assessment to use an accepted or appropriate measure.

Thank you for your suggestion. We went through all included animal studies to assess the risk of bias according to the SYRCLE RoB tool. Consequently, we added to the Methods: “Animal experiments were assessed for risk of bias based on the SYstematic Review Centre for Laboratory animal Experimentation (SYRCLE) RoB tool. ²³ Additionally, incomplete reporting of basic information relating to the study protocol was graded with a score from 0 (no selective reporting) to 20 (highest selective reporting) according to criteria listed in Supplementary Table S5.”. Corresponding results read as: “Overall, the majority of animal studies presented with unclear RoB, due to limited reporting on the items targeted by the SYRCLE tool. In particular, items corresponding to selection (sequence generation and allocation concealment), performance (random housing and blinding) and attrition (incomplete outcome data) biases were rated as unclear for 59.2%, 92.2%, 99.6%, 91.8% and 87.5% of the experiments included, respectively. An important other source of bias identified was the frequent use of additional drugs, including anesthetics, painkillers and antibiotics.”.

REVIEWERS' COMMENTS:

Reviewer #2 (Remarks to the Author):

Many thanks for making these minor amendments, I am satisfied that these have been suitably addressed in the revised manuscript.